# Age-Related Decline in Brain Myelination: Quantitative Macromolecular Proton Fraction Mapping, T2-FLAIR Hyperintensity Volume, and Anti-Myelin Antibodies Seven Years Apart

**DOI:** 10.3390/biomedicines12010061

**Published:** 2023-12-27

**Authors:** Marina Khodanovich, Mikhail Svetlik, Anna Naumova, Daria Kamaeva, Anna Usova, Marina Kudabaeva, Tatyana Anan’ina, Irina Wasserlauf, Valentina Pashkevich, Marina Moshkina, Victoria Obukhovskaya, Nadezhda Kataeva, Anastasia Levina, Yana Tumentceva, Vasily Yarnykh

**Affiliations:** 1Laboratory of Neurobiology, Research Institute of Biology and Biophysics, Tomsk State University, 36 Lenina Ave., Tomsk 634050, Russia; mihasv@gmail.com (M.S.); nav@uw.edu (A.N.); kmsra08@gmail.com (M.K.); tany_a@list.ru (T.A.); irinawasserlauf@mail.ru (I.W.); nadi-51@yandex.ru (N.K.); nastasio@yandex.ru (A.L.); mimizyana@gmail.com (Y.T.); 2Department of Radiology, University of Washington, 850 Republican Street, Seattle, WA 98109, USA; 3Laboratory of Molecular Genetics and Biochemistry, Mental Health Research Institute, Tomsk National Research Medical Center of the Russian Academy of Sciences, Tomsk 634014, Russia; susl2008@yandex.ru; 4Cancer Research Institute, Tomsk National Research Medical Center of the Russian Academy of Sciences, 12/1 Savinykh St., Tomsk 634009, Russia; afina.tsk@gmail.com; 5Department of Fundamental Psychology and Behavioral Medicine, Siberian State Medical University, 2 Moskovskiy Trakt, Tomsk 634050, Russia; 6Department of Neurology and Neurosurgery, Siberian State Medical University, 2 Moskovskiy Trakt, Tomsk 634050, Russia; 7Medica Diagnostic and Treatment Center, 86 Sovetskaya st., Tomsk 634510, Russia

**Keywords:** normal aging, myelin, quantitative MRI, neuroimaging, macromolecular fraction mapping, MPF, white matter, FLAIR hyperintensities, demyelination, myelin-specific autoantibodies

## Abstract

Age-related myelination decrease is considered one of the likely mechanisms of cognitive decline. The present preliminary study is based on the longitudinal assessment of global and regional myelination of the normal adult human brain using fast macromolecular fraction (MPF) mapping. Additional markers were age-related changes in white matter (WM) hyperintensities on FLAIR-MRI and the levels of anti-myelin autoantibodies in serum. Eleven healthy subjects (33–60 years in the first study) were scanned twice, seven years apart. An age-related decrease in MPF was found in global WM, grey matter (GM), and mixed WM–GM, as well as in 48 out of 82 examined WM and GM regions. The greatest decrease in MPF was observed for the frontal WM (2–5%), genu of the corpus callosum (CC) (4.0%), and caudate nucleus (5.9%). The age-related decrease in MPF significantly correlated with an increase in the level of antibodies against myelin basic protein (MBP) in serum (r = 0.69 and r = 0.63 for global WM and mixed WM–GM, correspondingly). The volume of FLAIR hyperintensities increased with age but did not correlate with MPF changes and the levels of anti-myelin antibodies. MPF mapping showed high sensitivity to age-related changes in brain myelination, providing the feasibility of this method in clinics.

## 1. Introduction

Identifying a pathological substrate of cognitive decline in normal aging and age-related dementias becomes increasingly important with the continuous growth of life expectancy. It is commonly recognized that normal brain aging is a multifaceted process primarily affecting the microstructure of neuropil while keeping the global neuronal population nearly intact [1,2]. Age-related decline in brain function suggests that disruption of normal brain connectivity may be an important underlying reason. This circumstance motivated substantial attention to investigations of myelin changes in the aging brain.

A number of histological studies in rodents [3] and non-human primates [4,5] have shown the effect of age on the ultrastructure of myelinated nerve fibers such as “balloons” in myelin sheaths [4,6], an increase in the number of lamellae [7], and accumulation of dense cytoplasm with vesicular inclusions [6]. Paler staining of white matter was observed in the brains of old primates compared to young animals [8]. Histological studies on the post-mortem human brain have also shown a decrease in the intensity of staining of nerve fibers in the cerebral cortex [9]. Stereological analyses [10] indicated a 15–17% age-related loss of white matter in the cerebral hemispheres and a 27% reduction in the total length of myelinated nerve fibers.

The mechanisms of WM degeneration in normal aging are mainly associated with changes in the state of oligodendrocytes (OL) and oligodendrocyte precursors (OPC). OL exhibit signs of degeneration before other cells in the brain due to high metabolic demands and the vulnerability to oxidative damage [11]. Age-related changes in the WM microenvironment, including the accumulation of myelin debris and the reactivation of astroglia and microglia, which release cytokines, pro-inflammatory, and cytotoxic factors, impair the ability of mature OL to support myelin maintenance and inhibit OPC recruitment and differentiation into myelinating OL [11,12,13].

Histological studies of age-related myelin changes in humans are rare, highly dependent on the quality of the biomaterial fixation, and can only be carried out post-mortem. An alternative approach that allows for longitudinal study of brain myelination is the use of non-invasive imaging. A number of magnetic resonance imaging (MRI) methods are sensitive to myelin, though their specificity varies greatly [14]. Nevertheless, modern neuroimaging studies collectively suggest a progressive decline in myelination in the aging brain. Diffusion imaging methods, and, particularly, diffusion tensor imaging (DTI), have been the most common approach to studying age-related changes in white matter (WM) [15,16,17,18,19,20,21]. DTI studies demonstrated age dependence of myelin-associated indices (fractional anisotropy (FA) and radial diffusivity (RD)) consistent with demyelination in various WM tracts, including frontal WM, internal capsule (IC), and corpus callosum (CC) [20,21]. A major limitation of DTI for myelin assessment is the sensitivity of FA and RD to the direction of diffusion in regions containing multidirectional fibers and changes in tissue microstructure unrelated to myelination [19]. Another, more myelin-specific technique, myelin water fraction (MWF) mapping, has shown an age-related decrease in myelination for most WM regions [17,18,20,21,22,23]. A semi-quantitative technique for studies of cortical myelination based on the signal ratio obtained from T1- and T2-weighted images revealed inverted U-shaped aging profiles in the cortex, suggesting protracted myelination in younger ages followed by a decline in elderly subjects [24]. Similar profiles were identified later for the age dependence of R1 = 1/T1 [25] and MWF [17] in WM. Quantitative interpretation of the above findings strictly in terms of myelin is difficult because changes in relaxation times T1 and T2 affect both measured signal intensities and MWF [26]. Such changes are non-specific to myelin and may be caused by age-related iron deposition, water content alterations, and gliosis [26,27].

The magnetization transfer (MT) effect provides an alternative mechanism for probing myelination changes via interaction between water and macromolecules in tissues [28,29]. In early brain aging studies (a detailed review can be found in [28], the MT effect was assessed by the empirical semi-quantitative index MT ratio (MTR). Several studies have shown negative associations between MTR and age, however, MTR provided lower sensitivity to age-related changes as compared with MWF and DTI [18]. The main limitation of MTR, which likely explains the lower specificity and sensitivity for myelin, is its dependence on longitudinal relaxation [30]. This dependence partially offsets the effect of macromolecular composition changes and additionally introduces unwanted sensitivity to paramagnetic effects. A newer MT quantitation approach, the MT saturation index (MTsat), is independent of relaxation and offers improved specificity and sensitivity to age-related changes in myelination [31,32,33,34]. An alternative and more comprehensive way to characterize the MT effect is based on quantitative mapping of the fundamental parameters of the two-pool MT model [35] using specialized quantitative MT (qMT) techniques [29]. While applications of qMT in aging research are limited to date, particular parameters of the two-pool model (specifically, the macromolecular proton fraction (MPF) and cross-relaxation rate constant) showed negative correlations with age at the whole-brain level and in certain WM structures [36].

Among the parameters of the two-pool model, MPF has attracted major interest as a biomarker of myelin [29]. Multiple studies in animal models evidenced strong correlations between MPF and myelin histology and high sensitivity of this parameter to myelin damage, as detailed in the recent review [29]. A recently emerged fast single-point MPF mapping method [37] allows the reconstruction of MPF maps in isolation from other two-pool model parameters and overcomes the key limitations of most qMT techniques related to extremely time-consuming data acquisition and high sensitivity to noise. The fast MPF mapping method [37] offers several practical advantages, including independence of magnetic field strength [38], insensitivity to major confounders of alternative myelin imaging methods, such as myelinated fiber orientation [19] and iron deposition [39], high reproducibility [37], and easy implementation based on routine MRI equipment without modification of the original manufacturers’ pulse sequences [40,41]. This method has been extensively validated by histology in the normal animal brain [38] and the animal models of cuprizone-induced demyelination and remyelination [42,43], ischemic stroke [44,45], and neonatal development [46].

Recent applications of fast MPF mapping in the studies of multiple sclerosis [39,47], mild traumatic brain injury (mTBI) [30], schizophrenia [40], and normal brain development in fetuses [41], children [48], and adolescents [49] have shown the feasibility of using this method as a robust and versatile tool for the quantitative assessment of demyelination and myelin development in humans. Previous studies also highlighted the capability of MPF mapping to measure GM demyelination in isolation from the confounding effect of iron [39] and sensitivity of MPF to subtle changes in the myelin content associated either with age (in adolescence [49]) or conditions that are not primarily demyelinating (mTBI [30] and schizophrenia [40]).

The primary goal of this study was the quantification of age-related changes in global and regional myelination of the normal adult human brain using the fast MPF mapping method. Unlike the previous neuroimaging studies where the effect of age on myelination was examined cross-sectionally at a single time point per subject, the present study is based on the longitudinal assessment of myelination in the same subjects with the seven-year time interval. Additionally, we investigated age-related changes in the concentrations of autoantibodies to major myelin proteins in blood plasma, which could serve as peripheral markers of myelin damage according to the literature [50,51,52,53] and white matter hyperintensities on T2-FLAIR images, which are commonly viewed as signs of focal demyelination [54].

## 2. Materials and Methods

### 2.1. Subjects

Eleven healthy volunteers participated in the study. The inclusion criteria were the following: age from 33 to 60 years, the absence of a history of traumatic brain injury, and the absence of any diagnosed neurologic or psychiatric condition. The exclusion criteria were: contra indications to MRI, inability to tolerate the MRI procedure, and self-withdrawal from the study. Written informed consent was obtained from all participants.

For the first MR imaging, participants were recruited from staff at the Mental Health Research Institute between April and November 2015. The second MR imaging was performed on the same participants between September and November 2022. The demographic characteristics of participants are shown in Table 1.

### 2.2. MRI Data Acquisition

All participants underwent the first (2015) and repeated (2022) MR imaging using the 1.5 T clinical scanner Magnetom Essenza (Siemens, Erlangen, Germany) with an 8-channel head coil. The MRI equipment did not undergo any software or hardware upgrades or major repairs during the study. The fast MPF mapping protocol [40] included three 3D spoiled gradient-echo pulse sequences with the following acquisition parameters:MT-weighted: TR = 20 ms, echo time (TE) = 4.76 ms, flip angle (FA) = 8°, scan time 5 min 40 s;T1-weighted: TR = 16 ms, TE = 4.76 ms, FA = 18°, scan time 4 min 32 s;Proton-density-weighted: TR = 16 ms, TE = 4.76 ms, FA = 3°, scan time 4 min 32 s.In addition, the following sequences were included in the protocol:3D FLAIR-SPACE-FSE: TR = 5000 ms, TE = 390 ms, TI = 1800 ms;3D T2-SPACE-FSE: TR = 3000 ms, TE = 335 ms.

MT-weighted, T1-weighted, and PD-weighted images were used for further MPF map reconstruction, as described previously [37]. T2-FLAIR images were used to examine focal lesions (WM hyperintensities) with additional control of lesion detection on T2-weighted images.

All scans were acquired in the sagittal plane with a voxel size of 1.25 × 1.25 × 1.25 mm^3^ (matrix 192 × 192 × 160, field of view 240 × 240 × 200 mm^3^), single signal averaging.

The total scanning time was about 35 min.

### 2.3. Image Processing

MPF maps were reconstructed using the previously developed software in the C++ language (available at https://www.macromolecularmri.org/ (assessed on 22 August 2022)), which implements a single-point algorithm with a synthetic reference image [37]. MPF maps obtained from the first and repeat MRI were examined for 7-year global and regional changes in WM and GM. In addition, age-related changes in the volume of T2-FLAIR hyperintensities and MPF values in the hyperintense areas were assessed.

Image processing was performed using freely available MRIcro (Version 1.40, [55], ImageJ (Fiji) (version 1.54f, National Institutes of Health, Bethesda, MD, USA) [56], ITK-snap (version 4.0.1, University of Pennsylvania, PA, USA) [57], FSL (version 6.0.1, Analysis Group, FMRIB, Oxford, UK) [58], and Advanced Normalization Tools (ANTs) (version 2.1.1, University of Pennsylvania, PA, USA) [59,60] software tools. MRIcro was used for skull stripping of MPF maps and viewing. Manual editing and binarization of brain masks were performed in ImageJ. FSL was used for the MPF map segmentation of global compartments of WM, GM, and mixed GM–WM. The Advanced Normalization Tools (ANTs) package was used for the registration of the Eve anatomical atlas [61] to individual MPF maps to obtain individual regional segmentation. ITK-snap [57] was used for the manual delineation of focal lesions on T2-FLAIR images and MPF measurements in all segmented regions.

The first stage of global WM and GM changes measurements on MPF maps involved skull stripping using a mask, which was obtained by applying the BET algorithm to the PD-weighted images in the MRIcro application. The mask was then converted to a binary image in ImageJ using the Threshold function and applied to the MPF maps to remove extracerebral tissue. Automatic global segmentation of MPF maps was performed using the FSL package to obtain masks of WM, GM, mixed WM–GM, and mixed GM with cerebrospinal fluid (CSF), as detailed earlier [40,47]. Since MPF maps are not affected by the coil reception profile and have negligible sensitivity to B_1_ field inhomogeneity (particularly at 1.5 T [61]), bias field correction was not used. Masks were used to measure the mean MPF values of these compartments. The measurements were carried out using the ITK-SNAP application.

Regional WM and GM segmentation was carried out using Advanced Normalization Tools (ANTs) [59,60] and Eve anatomical atlas [62]. The T1 template image of the Eve atlas was registered to individual MPF maps using the antsRegistrationSyNQuick algorithm. Then, the obtained deformation field was applied to Type-III Eve atlas segmentation [62] to register the template atlas labels to individual MPF maps (Figure 1).

The measurements on MPF maps were performed for 118 GM and WM structures of the right and left hemispheres using ITK-snap software. The list of structures included:Juxtacortical (superficial) WM: superior parietal, superior, middle, and inferior frontal; precentral; postcentral; angular; pre-cuneus; cuneus; lingual; fusiform; superior, inferior, and middle occipital; superior, inferior, and middle temporal; lateral and middle fronto-orbital, supramarginal, rectus, cingulum (parts of cingulate gyrus and hippocampus);WM pathways and fasciculi: corticospinal tract (CST); medial lemniscus; anterior limb, posterior limb, and retrolenticular part of internal capsule (IC); inferior, superior, and middle cerebellar peduncles (CP); cerebral peduncles; posterior thalamic radiation; anterior, superior, and posterior corona radiata (CR); fornix (FX) (stria terminalis, column, and body); superior longitudinal (SL) fasciculus; superior (SFO) and inferior fronto-occipital (IFO) fasciculi; uncinate fasciculus; sagittal stratum; external capsule; pontine crossing tract; genu, body, and splenium of corpus callosum (CC); tapatum;Subcortical and allocortical GM structures: amygdala; hippocampus; entorhinal area; caudate nucleus; putamen; globus pallidus; thalamus;Brainstem structures: midbrain; pons; medulla.

The measurements for the left and right hemispheres were averaged for all brain structures except for juxtacortical WM. A series of juxtacortical WM labels corresponding to specific gyri were analyzed separately from other WM structures, taking into account whether they belonged to the left or right hemisphere.

WM hyperintensities were outlined manually by two operators blinded to the subject information and scan time point on T2-FLAIR images with the guidance of T2-weighted images. Then, T2-FLAIR images were registered to MPF maps using ITK-snap software to measure mean MPF values in the outlined areas.

### 2.4. ELISA

Whole blood samples from each subject were taken from the cubital vein after 12-h overnight fasting and centrifuged for 20 min at 2000× *g* at 4 °C. The serum was isolated and stored at −80 °C. Quantitative analyses of IgG antibodies against myelin basic protein (MBP) and proteolipid protein (PLP) in the serum were executed using respective ELISA kits by Cloud-Clone Corp. (CCC, Houston, TX, USA) according to the manufacturer’s instructions. The absorbance of each well at 450 nm was measured on a Varioskan LUX spectrophotometer (Thermo Scientific, Waltham, MA, USA) located at the core facility Medical Genomics at Tomsk National Research Center. Measurement results are presented in terms of optical density units (ODU).

### 2.5. Statistical Analysis

Statistical analysis was performed using Statistica 10.0 software. Differences between time points, inter-hemispheric, and gender differences were analyzed using the repeated measures analysis of variance (ANOVA) followed by post-hoc Fisher LSD tests. In the analyses of multiple brain structures, *p*-values were adjusted using the Benjamini–Hochberg procedure for false discovery rate (FDR) correction to prevent false positive results in multiple comparisons. FDR level was set to 0.05. Associations between variables were assessed using the Pearson correlation coefficient. Statistical significance for all analyses was taken as less than 0.05.

## 3. Results

### 3.1. Age-Related Global Changes in the Brain Myelination

Figure 2 demonstrates global changes in MPF and volumes of WM, GM, and mixed WM–GM compartments. Age-related decrease in myelination was significant in WM, GM, and mixed WM–GM (Figure 2b). However, the percentage MPF decrease in WM (1.30 ± 0.55%) was twice greater than that in GM (0.75 ± 0.72%) and mixed WM–GM (0.75 ± 0.72%) (Figure 2c). A decrease in volume was significant only for GM, with a relative change of 2.6% (Figure 2d). No gender differences in MPF were found for both WM, mixed WM–GM, and GM.

### 3.2. Age-Related Changes in Separate WM and GM Structures

Figure 3 shows age-related MPF differences in the explored WM and GM structures in detail. Additionally, Figure 3a highlights interhemispheric differences in MPF for juxtacortical WM for both time points. Interhemispheric differences, with a few exceptions, coincide for 2015 and 2022. In the frontal lobe, the MPF was higher in the right hemisphere compared to the left one (differences were significant for the rectus, middle frontal, lateral and middle fronto-orbital cortical WM for both time points, for precentral WM—only for 2015). Higher MPF values in the right hemisphere were also found for the fusiform WM for both time points. For the parietal occipital lobes and cingulum, on the contrary, the MPF values were higher in the left hemisphere (differences are significant for pre-cuneus, cuneus, supramarginal WM for both time points, for lingual, occipital, and hippocampal part of cingulum WM—only for 2022). Temporal juxtacortical WM did not show significant interhemispheric differences.

In almost all structures of both WM and GM, a decrease in MPF was observed, but not in all structures was it statistically significant. All frontal juxtacortical WM areas showed significant MPF decrease both in the left and right hemispheres (except for precentral WM, significant differences were only in the right hemisphere), whereas lateral occipital WM did not show a significant decrease over 7 years (Figure 3a). Lateral temporal, precentral, and postcentral WM showed significant differences, mostly in the right hemisphere.

The medial and inferior parts of juxtacortical WM revealed some heterogeneity of age-related changes. The cingulate part of the cingulum, fusiform, and rectus WM showed a significant decrease in both hemispheres, whereas in the cuneus, pre-cuneus, and hippocampal parts of the cingulum, WM differences were not observed. Lingual WM significantly decreased only in the left hemisphere. A significant decrease in MPF was also found for the majority of the investigated WM pathways (Figure 3c), including CC (except for the splenum), posterior thalamic radiations, anterior and superior CR, SL, uncinate, SFO and IFO fasciculi, anterior and posterior limbs of IC, external capsule, column and body of FX, sagittal stratum, and tapatum. No differences were found in the cerebellar pathways, cerebral peduncles, pontine crossing tract, stria terminalis, posterior part of CR, retrolenticular part of IC, and splenum of CC. In addition, MPF significantly decreased in all investigated GM and mixed WM–GM structures (Figure 3b), including all basal ganglia, thalamus, hippocampus, amygdala, entorhynal area, and brainstem (except for the pons). No significant gender effect on the amount of the MPF decrease was found.

The percentage changes between the mean MPF values obtained in 2015 and 2022 for the same brain structures are shown in Figure 4. The most apparent decrease in MPF in the juxtacortical WM (Figure 4a), by 1.8–5.4%, was observed in the frontal lobe. In other lobes of the brain, the decrease in MPF was much smaller. Only for fusiform and singular juxtacortical WM, the decrease reached 2% in both hemispheres. The same magnitude of a decrease in MPF was found for the precentral and central, as well as inferotemporal juxtacortical WM of the right hemisphere. Among the WM tracts (Figure 4b), the genu of CC achieved the largest decrease of 4%. Anterior WM regions (anterior CR, anterior limb of IC, genu of CC) generally showed a higher percentage reduction in MPF (exceeding 2%) as compared to posterior WM regions. In the segmented tracts, MPF reductions were mostly significant, with the largest effect in the uncinate fasciculus (2.8%). Among all brain regions studied, the greatest decrease of 5.9% was observed for the caudate nucleus (Figure 4c). In other deep GM structures, the differences were smaller and amounted to 1.6% for the globus pallidus, 1.9% for the putamen, and 2.2% for the thalamus. Allocortical structures showed a decrease of around 2% (2.4% for the amygdala, 2.0% for the hippocampus, and 2.2% for the entorhinal area). MPF in the brainstem structures decreased to the smallest extent (1.3% for midbrain, 1.2% for medulla, and 0.6% for pons).

Figure 5 summarizes our findings and shows regions of a significant age-related decrease in MPF in representative cross-sections of a 3D MPF map.

### 3.3. Age-Related Changes in the Volume of T2-FLAIR Hyperintensities

Typical age-related changes of the brain reflected in T2-FLAIR images and MPF maps are shown in Figure 6. Specifically, zones of hyperintensities were detected on images obtained in 2022; those hyperintensity zones were mainly located periventricularly (Figure 6a). Over 7 years, the volume of hyperintense zones increased by an average of 35.2% (Figure 6b), both due to an increase in the volume of lesions adjacent to the lateral ventricles and due to the appearance of new lesions in bulk WM. (Figure 6a). Average MPF within T2-FLAIR hyperintensities slightly decreased, but the differences between 2015 and 2022 were not statistically significant (Figure 6b).

The percentage changes in the volume of T2-FLAIR hyperintensities between men and women were close to significant (*p* = 0.053). In men, the percentage changes in the volume of hyperintensities from 2015 to 2022 was 20.3 ± 20.0% on average, while in women, the average changes reached 50.0 ± 24.4%.

### 3.4. Age-Related Changes in Myelin-Related Autoantibodies

In 2022, the levels of antibodies against MBP tended to increase compared to 2015 (0.146 ± 0.109 ODU in 2015, 0.158 ± 0.154 ODU in 2022), but this difference (8.8 ± 1.6%) was not statistically significant. The levels of antibodies against PLP in 2022 and 2015 were nearly identical (0.042 ± 0.007 ODU in 2015, 0.043 ± 0.006 ODU in 2022, percentage changes 1.3 ± 0.8%). No gender differences were found for both antibodies.

Significant negative correlations were found between the percentage differences in anti-MBP antibody concentration and the percentage differences in global WM (r = 0.69, *p* < 0.05) and mixed WM–GM (r = 0.63, *p* < 0.05). The correlation of percentage changes in anti-MBP antibodies with the percentage changes in the volume of WM hyperintensities was insignificant. No significant correlations were found for the age-related percentage differences in antibodies to PLP.

## 4. Discussion

This study, for the first time, has demonstrated an age-related decrease in WM and GM myelination measured by quantitative MPF mapping. Specifically, a significant decrease in MPF between repeated MRI scans seven years apart was found for both global WM and mixed WM–GM compartments, as well as for the majority of WM and GM structures studied. The study was conducted on the same subjects using the same MRI scanner, which previously has shown highly reproducible results [40].

To the best of our knowledge, no other longitudinal studies have been published so far describing age-related changes in myelin content of the same subjects examined in two consecutive scans several years apart. Typically, the publications of age-related alterations in brain myelination report the results of cross-sectional studies on a fairly large sample of healthy subjects of different ages [15,16,17,18,22,23,33,63,64,65]. Therefore, our current study is unique in this aspect.

Several earlier quantitative neuroimaging studies reported findings consistent with the decline in myelin content during normal aging. Particularly, DTI studies showed age-related changes in FA and RD in various WM regions and fiber tracts [15,16,66]. O’Sullivan et al. [16] reported an age-related decrease in FA in the frontal WM. In the work of Salat et al. [15], age-related FA decrease was found only in the frontal WM, the posterior limb of the IC, and the genu of the CC out of 9 studied brain areas; the FA in the temporal and posterior WM was relatively preserved. Most recent publications [17,18] reported a stronger correlation of MWF with age but weaker correspondence to age of FA and RD parameters. Arshad et al. [17] investigated the relationship of DTI and MWF parameters with age within two models—linear and quadratic (inverted-U). It was found that FA and RD correlate with age only within the linear model, while MWF correlates with age within the inverted-U (quadratic) model, which better describes the age–myelin association in a wide range of ages [17,66]. Similar results were reported by Faizy et al. [18]. Specifically, it was found that MWF significantly decreased with age in most WM regions (except corticospinal tract). FA and MTR were associated with lower MWFs in the commissural fiber tracts only; RD had no regional effects on MWF. The authors concluded that DTI and MTI methods have limited specificity to myelin. Other publications that used MTI and MTR methods for the evaluation of age-dependent myelin changes [18,33,64] also showed weak dependence of MTR from age.

Our studies of myelination using MPF mapping to assess myelination are consistent with the above findings regarding myelin decline with age.

Our results highlight the capability of fast MPF mapping to capture very subtle changes in brain myelination with the use of a relatively small sample size. This feature is a direct consequence of the high precision of the fast single-point MPF mapping method. Earlier studies demonstrated excellent scan–rescan repeatability of MPF measurements in brain tissues, confirmed by small within-subject coefficients of variation (CV) in ranges of 1–2% in animals [42] and 0.8–1.6% in humans [40,61]. Of particular importance, the recent study performed on the same MRI unit with the same imaging protocol and global segmentation pipeline [40] reported the coefficients of variations of 0.8% in WM and 1.0% in GM in a short-term scan–rescan setting. These results corroborate our findings of a significant decrease in MPF by 0.75% in GM and 1.3% in WM. In summary, the current study affirms the feasibility of tracing a small decline in the myelin content associated with normal brain aging, which has a translation potential to detect larger effects of myelin disruption in various neurological conditions at the individual level.

Unlike MTR, MPF does not depend on T1 in tissues [37], which is especially important for studying age-related changes in myelin due to iron accumulation in the older brain [34]. Our recent study on patients with multiple sclerosis (MS) proved that MPF is an iron-insensitive measure of demyelination [39]. Our current study applied MPF to longitudinal repetitive measurements of myelin in the brains of healthy adults. A significant global decrease of 1.3% (*p* < 0.001) over seven years was found in WM myelination in the same subjects. A two-fold smaller but statistically significant decrease in MPF was also shown for GM (0.75% change, *p* < 0.05) and mixed WM–GM (0.75% change, *p* < 0.01) compartments. Similar to other studies [15,16,23,33], we found an anterior-to-posterior gradient in brain myelination changes. The greatest regional age-related changes in WM, reaching 5% over seven years, were detected in the frontal cortical WM. A decrease in myelin content was less pronounced but statistically significant in the parietal and temporal cortical WM and in the majority (15 of 26) of subcortical WM regions. We also showed an age-related myelination decrease in the mixed WM–GM brainstem structures (except the pons), and all explored allocortical and deep GM structures. Myelination decrease was most significant in the caudate putamen, 2–6% over seven years. To our knowledge, an age-related decrease in GM myelination has not been previously reported.

It is important to outline that the decrease in MPF in global WM and mixed WM–GM brain compartments significantly correlated with increased levels of MBP autoantibodies and major myelin proteins in the same subjects. Studies of autoantibodies of myelin-specific proteins in serum have mainly been carried out in the context of the hypothesis of their participation in the pathogenesis of multiple sclerosis [50,51,52,53]. The presence of antibodies to MBP was detected both in the serum of patients with MS and in the serum of healthy controls, although in lower concentrations [50,53]. For example, the study by Vojdani et al. [50], conducted with a control group of similar age (32–48 years), detected similar levels of antibodies to MBP, as was shown in our current study, in contrast to its threefold increase in MS patients. Greer et al. [52] provided more compelling evidence for the involvement of anti-PLP antibodies in the pathogenesis of MS, showing the association of these antibodies with clinical scores and disease severity. In the same study, the level of anti-PLP antibodies in healthy controls was extremely low, which is similar to our results. To the best of our knowledge, no studies investigated anti-myelin plasma antibodies in normal aging.

We did not find correlations between age-related changes in myelin antibodies and changes in the volume of T2-FLAIR hyperintensities, although the volume of those hyperintensity zones in WM increased significantly with age. MPF values within these lesions were substantially lower than those in the surrounding WM, thus suggesting the loss of myelin. Based on the literature, T2-FLAIR hyperintensities often correspond to the areas of demyelination but may also represent tissue damage associated with disruption of the structure of the extracellular matrix and the outflow of tissue fluids [54]. Such foci of hyperintensity frequently occur in elderly healthy people [67] and are arguably considered risk factors for stroke and dementia [68]. Although the T2-FLAIR sequences are typically used for identifying demyelination foci in clinics, studies showed that hyperintensity of the T2-FLAIR signal cannot be used for unambiguous detection of myelin damage and, especially, its quantitative assessment [54]. Our results indicate that visible abnormalities of the T2-weighted signal cannot explain the global trend of age-related myelin loss, which is widespread in radiologically normal brain tissue.

The discovered interhemispheric differences in cortical WM are particularly interesting. These differences were found independently for most brain regions in different years of the study. In other words, MPF maps obtained in 2015 and 2022 were not registered to each other but were registered to the brain atlas separately.

For example, MPF in the frontal WM (rectus, precentral, middle frontal, lateral, and middle fronto-orbital WM) was significantly higher in the right hemisphere; however, MPF in the parietal and occipital lobe (pre-cuneus, cuneus, supramarginal, lingual, and occipital WM) was higher for the left hemisphere. The exception is fusiform WM, where myelination is higher in the right hemisphere. Age-related changes were associated with interhemispheric differences in more regions of juxtacortical WM: in 2022, unlike in 2015, interhemispheric differences were found in the lingual, occipital, and hippocampal part of cingulum WM, whereas only the precentral WM showed interhemispheric differences in 2015, unlike 2022.

Previously published studies of age-related changes in brain myelination are contradictory. Faizy et al. [18] did not find interhemispheric differences in the frontal, parietal, and occipital WM and CST despite the use of several measurement methods (MWF, FA, MD, RD, and MTR). Other studies that used the MTR method to assess myelination have found larger values for this parameter [64] and large age-related MTR differences in the left hemisphere [33]. MPF mapping and detailed segmentation used in the present study were able to find consistent interhemispheric differences in juxtacortical WM myelination and their dependence on age.

Since many studies consider WM loss as the basis of cognitive decline in normal aging [13,16,63,69,70,71], it would be of great interest to investigate associations of demyelination with performance on cognitive tests. Future studies will define the contribution of demyelination of specific brain structures to age-related decline of cognitive abilities.

## 5. Conclusions

This is the first longitudinal study of age-related changes in brain myelination assessed with quantitative MPF mapping on the same subjects seven years apart. A significant age-related decrease in MPF was found in global WM and GM, as well as in 48 out of 82 WM and GM regions. The most pronounced MPF decrease was observed for the frontal juxtacortical WM (2–5%), genu of CC (4.0%), and caudate nucleus (5.9%). The smallest age-related changes in MPF were found in the brainstem (0.6–1.3%). The age-related decrease in MPF significantly correlated with an increase in the level of anti-MBP antibodies in the blood plasma. The volume of WM hyperintensities increased with age but did not correlate with either changes in MPF or the concentration of myelin-specific antibodies.

Our results demonstrate the high sensitivity of MPF to age-related changes in brain myelination, which confirms the feasibility of longitudinal studies based on the fast MPF mapping method in a clinical setting. Information about longitudinal MPF changes during normal aging provides a methodological background for future neuroimaging studies of myelin in age-related brain diseases.

## 6. Study Limitations

The study was conducted on a small sample and a small age range of the studied subjects. Age-related decline in myelination has not been studied for subjects older than 67 years. The study did not include independent control samples of similar age at both time points to carry out cross-sectional comparisons. Statistical analysis was performed within a linear model, whereas age-related changes in myelin were shown to be more consistent with a quadratic (inverted-U) model. However, a quadratic model would be less suitable for our study due to a narrow range of ages and a small sample size.

## Figures and Tables

**Figure 1 biomedicines-12-00061-f001:**
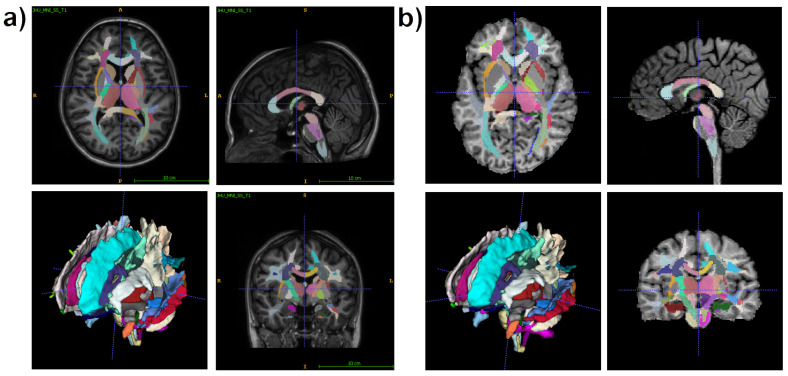
An example of T1 Eve template [62] registration with segmentation (**a**) to individual MPF map (**b**). Slices are shown in similar axial, sagittal, and coronal projections. R-L—right to left, S-I—superior to inferior.

**Figure 2 biomedicines-12-00061-f002:**
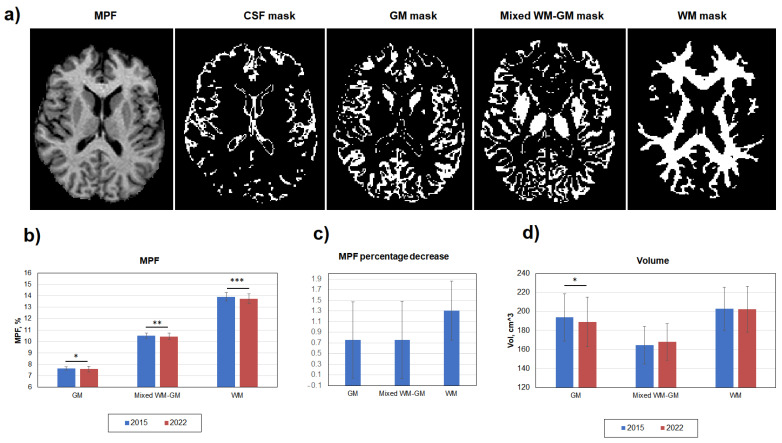
Age-related global changes of brain myelination for WM, GM, and mixed WM–GM compartments. (**a**) Example MPF map and corresponding masks of cerebrospinal fluid (CSF), GM, WM, and mixed WM–GM used for global measurements. (**b**) Absolute MPF decrease in global GM, WM, and mixed WM–GM. (**c**) Percentage MPF decrease in global GM, WM, and mixed WM–GM. (**d**) Volume changes in GM, WM, and mixed WM–GM. Error bars denote standard deviation. Significant differences: *—*p* < 0.05, **—*p* < 0.01, ***—*p* < 0.001.

**Figure 3 biomedicines-12-00061-f003:**
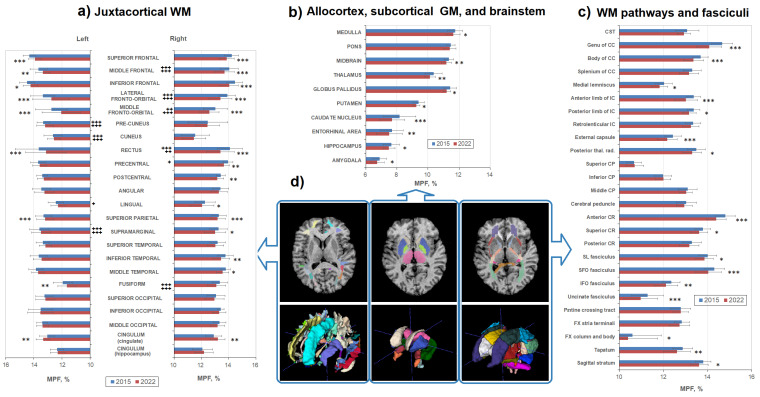
Age-related changes between MPF measurements obtained in 2015 and 2022 for the separate brain regions: (**a**) juxtacortical WM of left and right hemispheres, (**b**) allocortex, subcortical GM, and brainstem, and (**c**) WM pathways. (**d**)—Example segmentation on an individual MPF map (upper row) and 3D reconstruction of three sets of brain structures corresponding to (**a**–**c**) measurements. Colors correspond to the labels of separate brain structures. Significant differences between 2015 and 2022: *—*p* < 0.05, **—*p* < 0.01. ***—*p* < 0.001. Significant differences between left and right hemispheres: +—*p* < 0.05, ++—*p* < 0.01. +++—*p* < 0.001. The significance of the differences is marked on the side of the hemisphere in which the MPF is larger. Error bars correspond to SD.

**Figure 4 biomedicines-12-00061-f004:**
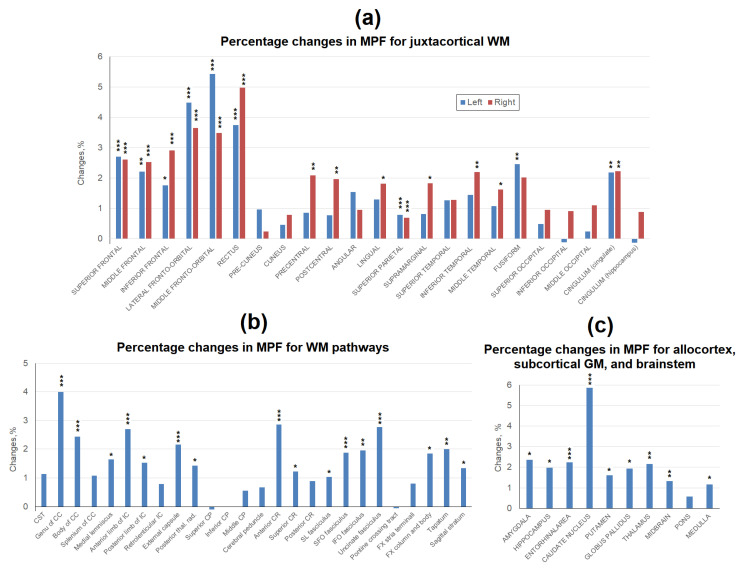
Age-related percentage changes between average MPF measurements obtained in 2015 and 2022 for the separate brain regions: (**a**) juxtacortical WM of left and right hemispheres, (**b**) WM pathways, (**c**) allocortex, subcortical GM, and brainstem. Significant differences between 2015 and 2022: *—*p* < 0.05, **—*p* < 0.01. ***—*p* < 0.001.

**Figure 5 biomedicines-12-00061-f005:**
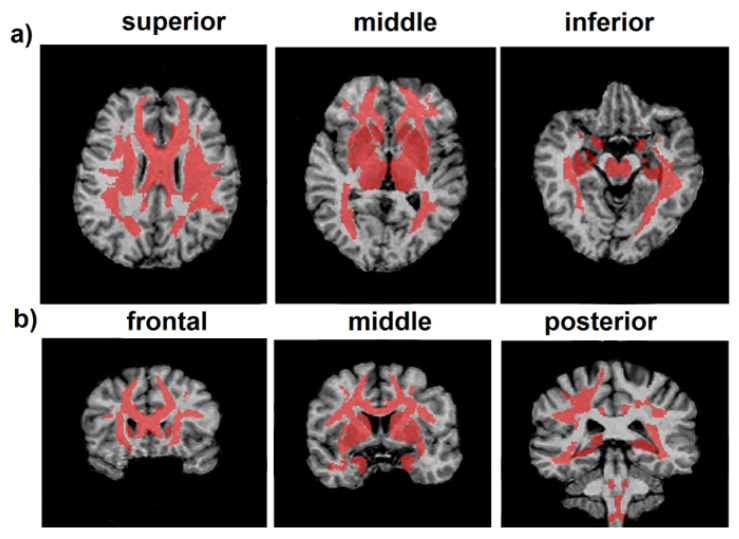
Regions of a significant age-related decrease in MPF in representative cross-sections of a 3D MPF map at different levels of axial (**a**) and coronal (**b**) projections of an individual MPF map. Regions of significant MPF decrease are marked by red color.

**Figure 6 biomedicines-12-00061-f006:**
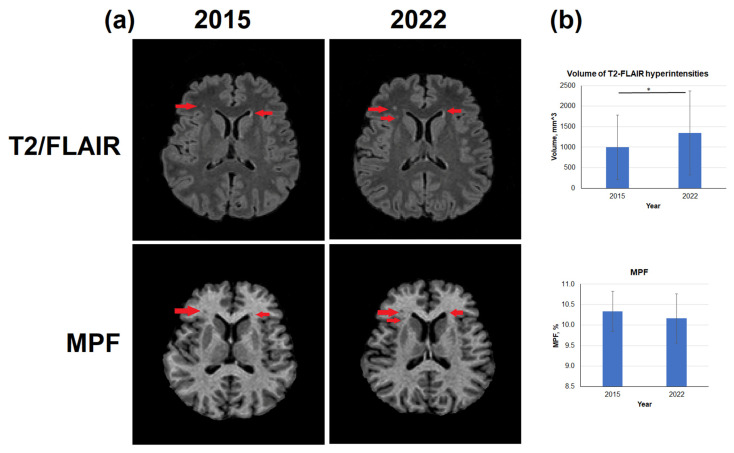
Age-related changes in the volume of T2-FLAIR hyperintensities and MPF in hyperintense zones. (**a**)—Example T2-FLAIR images and MPF maps obtained from the same subject in 2015 and 2022. T2-FLAIR hyperintensities correspond to MPF hypointensities are marked by red arrows. (**b**)—Quantitative differences in the volume of T2-FLAIR hyperintensities and MPF in it. Significant differences: *—*p* < 0.05. Error bars correspond to SD.

**Table 1 biomedicines-12-00061-t001:** The demographic characteristics of participants of the study. SD—standard deviation.

Parameter	Total	Male	Female
Sample size (%)	11 (100)	6 (55)	5 (45)
Age, 2nd study, years (SD)	52.2 (8.6)	54.7 (9.1)	49.2 (7.7)
Age, 1st study, median (min–max)	44 (33–60)	48.5 (36–60)	41 (33–54)
Age, 2nd study, median (min–max)	51 (40–67)	55.5 (43–67)	48 (40–61)

## Data Availability

Data are unavailable due to privacy or ethical restrictions.

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
