# Peer review of "Age-Related Decline in Brain Myelination: Quantitative Macromolecular Proton Fraction Mapping, T2-FLAIR Hyperintensity Volume, and Anti-Myelin Antibodies Seven Years Apart"

_biomedicines, 2023, doi:10.3390/biomedicines12010061_

Round 1
Reviewer 1 Report
Comments and Suggestions for Authors
Appraisal: In the introduction, the role of glia in the aging and related processes is not clear. For example, there is no difference between astrocytes, oligodendrocytes, and aging. Indeed, this information is crucial and needs to be well explained and defined in the introduction. Despite the interest for the decline in the human brain, several studies investigated the relationship between glial cells and aging in animal models.
About DTI, the authors need to clarify if they mean DWI or DTI. Indeed, DTI is the mathematical model underlying DWI. This difference needs to be specified in the introduction.
I like the description about MPF, but the authors described the application (i.e. aging, multiple sclerosis etc.) and they should add details about the results obtained in previous studies.
Authors stated in the aims “Additionally, we investigated age-related changes in the concentra-120 tions of autoantibodies to major myelin proteins in blood plasma, which served as periph-121 eral markers of myelin damage and white matter hyperintensities on T2-FLAIR images, 122 which are commonly viewed as signs of focal demyelination” but they did not reported a previous study that implemented a similar protocol. Similarly, they did not report the relationship between peripheral changes (which?) and brain myelin alterations.
The sample size is very small to perform a study about the age-related changes in myelinization.
Despite the small sample size, the study is interesting since the 7 years follow-up. Did the authors try to perform a comparison with two age-matched groups (one similar to the 2015 group and one similar to 2022 group)? A mixed design model (longitudinal + transversal ) can be useful to control time-related effect.
Line 138- “The MRI equipment did not undergo any software and hardware upgrades or major repairs during the study.” How is it possible? I can understand about hardware, but I do not believe about no software updates.
Please add the number of the channels of the coil (8 or 16?)
Statistical Analysis: Authors need to apply a multivariate model of rmANOVA. A simple rm ANOVA is not helpful to control the results for multiple comparisons. I also suggest to add IgG antibodies values as a covariate.
Despite some limitation, the results and the discussion are interesting and well written.
Author Response
We thank the Reviewer for valuable comments and critical evaluation.
- Appraisal: In the introduction, the role of glia in the aging and related processes is not clear. For example, there is no difference between astrocytes, oligodendrocytes, and aging. Indeed, this information is crucial and needs to be well explained and defined in the introduction. Despite the interest for the decline in the human brain, several studies investigated the relationship between glial cells and aging in animal models.
Answer: We thank to the reviewer for the important comments on the mechanisms of age-related myelin degeneration. The following paragraph has been added to Introduction:
The mechanisms WM degeneration in normal aging are mainly associated with changes in the state of oligodendrocytes (OL) and oligodendrocyte precursors (OPC). OL exhibit signs of degeneration before other cells in the brain due to high metabolic demands and the vulnerability to oxidative damage (Chapman and Hill 2020). Age-related changes in WM microenvironment, including accumulation of myelin debris, reactivation of astroglia and microglia, which release cytokines, pro-inflammatory and cytotoxic factors, impair the ability of mature OL to support myelin maintenance and inhibit OPC recruitment and differentiation into myelinating OL (Chapman and Hill 2020; Kohama, Rosene, and Sherman 2012; Sams 2021).
- About DTI, the authors need to clarify if they mean DWI or DTI. Indeed, DTI is the mathematical model underlying DWI. This difference needs to be specified in the introduction.
Answer: There is a variety of diffusion imaging methods, and DTI is one of them (probably, the most commonly used). In the context of the text fragment, we specifically refer to the results obtained with DTI. We rephrased the corresponding sentence.
- I like the description about MPF, but the authors described the application (i.e. aging, multiple sclerosis etc.) and they should add details about the results obtained in previous studies.
Answer: We added highlights of the previous studies relevant to the topic of the manuscript.
- Authors stated in the aims “Additionally, we investigated age-related changes in the concentrations of autoantibodies to major myelin proteins in blood plasma, which served as peripheral markers of myelin damage and white matter hyperintensities on T2-FLAIR images, which are commonly viewed as signs of focal demyelination” but they did not reported a previous study that implemented a similar protocol. Similarly, they did not report the relationship between peripheral changes (which?) and brain myelin alterations.
Answer: We apologize if the above sentence can be understood ambiguously. We meant the literature data suggesting that blood markers may reflect myelin damage in demyelinating diseases (mainly multiple sclerosis) and unintentionally omitted references. We rephrased the sentence of concern and added corresponding references as follows:
Additionally, we investigated age-related changes in the concentrations of autoantibodies to major myelin proteins in blood plasma, which could serve as peripheral markers of myelin damage according to the literature (Berger et al. 2003; Greer, Trifilieff, and Pender 2020; Hedegaard et al. 2009; Vojdani, Vojdani, and Cooper 2003), and white matter hyperintensities on T2-FLAIR images, which are commonly viewed as signs of focal demyelination (Haller et al. 2014).
The results related to blood markers and FLAIR image analysis are reported in sections 3.3 and 3.4.
- The sample size is very small to perform a study about the age-related changes in myelinization. Despite the small sample size, the study is interesting since the 7 years follow-up. Did the authors try to perform a comparison with two age-matched groups (one similar to the 2015 group and one similar to 2022 group)? A mixed design model (longitudinal + transversal ) can be useful to control time-related effect.
Answer: We thank the reviewer for the interesting idea. Unfortunately, we are not able to recruit and scan additional subjects due to cost limitations. We agree that the absence of independent control groups is a limitation of our study. We added the corresponding statement to the section Study Limitations.
- Line 138 “The MRI equipment did not undergo any software and hardware upgrades or major repairs during the study.” How is it possible? I can understand about hardware, but I do not believe about no software updates. Please add the number of the channels of the coil (8 or 16?)
Answer: The warranty period and initial service contract for this scanner with the manufacturer expired more than 10 years ago. Since then, the scanner has been maintained under the service contract with a local medical equipment company, which does not offer routine software upgrades. 8-channel head coil was used. We added this information to Methods.
- Statistical Analysis: Authors need to apply a multivariate model of rmANOVA. A simple rm ANOVA is not helpful to control the results for multiple comparisons. I also suggest to add IgG antibodies values as a covariate.
Answer: According to the Reviewer’s advice we have checked the differences between 2015 and 2022 using multivariate Repeated Measures model (the table below) and IgG as covariate. The results are similar to those initially presented using F-criteria with the Benjamini–Hochberg False Discovery Rate correction. All significant differences have been confirmed. The use of IgG as a covariate did not improve the model used.
|
Between 2015 and 2022 |
|
||
Brain structures |
Wilks lambda |
p |
|
|
Allocortex |
|
|
|
|
AMYGDALA |
0.578 |
0.022* |
|
|
HIPPOCAMPUS |
0.654 |
0.044* |
|
|
ENTORHINAL AREA |
0.346 |
0.001** |
|
|
Deep GM |
|
|
|
|
CAUDATE NUCLEUS |
0.244 |
0.000 |
|
|
PUTAMEN |
0.566 |
0.019* |
|
|
GLOBUS_PALLIDUS |
0.447 |
0.005** |
|
|
THALAMUS |
0.080 |
0.0000 |
|
|
Brainstem |
|
|
|
|
MIDBRAIN |
0.513 |
0.011* |
|
|
PONS |
0.856 |
0.224 |
|
|
MEDULLA |
0.525 |
0.013* |
|
|
WM pathways |
|
|
|
|
CST |
0.801 |
0.133 |
|
|
Inferior cereb ped |
0.999 |
0.981 |
|
|
Medial lemniscus |
0.607 |
0.029* |
|
|
Superior cereb ped |
0.987 |
0.727 |
|
|
Cerebral peduncle |
0.935 |
0.426 |
|
|
Anterior limb IC |
0.488 |
0.008** |
|
|
Posterior limb IC |
0.513 |
0.011* |
|
|
Posterior thal rad |
0.556 |
0.017* |
|
|
Anterior CR |
0.312 |
0.008** |
|
|
Superior CR |
0.602 |
0.028* |
|
|
Posterior CR |
0.780 |
0.124 |
|
|
FX Stria terminalis |
0.894 |
0.302 |
|
|
Superior longitudinal FAS |
0.662 |
0.047* |
|
|
Superior fronto-occipital FAS |
0.475 |
0.007** |
|
|
Inferior fronto-occipital FAS |
0.559 |
0.018* |
|
|
Sagittal stratum |
0.614 |
0.031* |
|
|
ExC |
0.573 |
0.021* |
|
|
Uncinate FAS |
0.651 |
0.431* |
|
|
Pontine crossing tract |
0.999 |
0.941 |
|
|
Middle cereb ped |
0.940 |
0.444 |
|
|
FX (column and body) |
0.650 |
0.430* |
|
|
Genu CC |
0.227 |
0.000* |
|
|
Body CC |
0.327 |
0.001** |
|
|
Splenium CC |
0.651 |
0.431* |
|
|
Retrolenticular IC |
0.882 |
0.274 |
|
|
Tapetum |
0.528 |
0.013* |
|
|
|
|
|
|
|
Juxtacortical WM |
Between 2015 and 2022, left hemisphere |
Between 2015 and 2022, right hemisphere |
||
|
Wilks lambda |
p |
Wilks lambda |
p |
SUPERIOR PARIETAL |
0.275 |
0.000*** |
0.276 |
0.000*** |
SUPERIOR FRONTAL |
0.409 |
0.003** |
0.451 |
0.006** |
MIDDLE FRONTAL |
0.566 |
0.012* |
0.502 |
0.014* |
INFERIOR FRONTAL |
0.612 |
0.033* |
0.336 |
0.001** |
PRECENTRAL |
0.866 |
0.241 |
0.351 |
0.001** |
POSTCENTRAL |
0.926 |
0.392 |
0.560 |
0.018* |
ANGULAR |
0.681 |
0.056 |
0.844 |
0.205 |
PRE-CUNEUS |
0.861 |
0.234 |
0.998 |
0.914 |
CUNEUS |
0.972 |
0.612 |
0.981 |
0.677 |
LINGUAL |
0.879 |
0.266 |
0.622 |
0.044* |
FUSIFORM |
0.330 |
0.001** |
0.276 |
0.000*** |
SUPERIOR OCCIPITAL |
0.949 |
0.482 |
0.895 |
0.305 |
INFERIOR OCCIPITAL |
0.997 |
0.871 |
0.896 |
0.307 |
MIDDLE OCCIPITAL |
0.984 |
0.695 |
0.778 |
0.122 |
SUPERIOR TEMPORAL |
0.726 |
0.081 |
0.671 |
0.051 |
INFERIOR TEMPORAL |
0.790 |
0.134 |
0.627 |
0.035* |
MIDDLE TEMPORAL |
0.812 |
0.159 |
0.606 |
0.028* |
LATERAL FRONTO-ORBITAL |
0.547 |
0.015* |
0.545 |
0.016* |
MIDDLE FRONTO-ORBITAL |
0.461 |
0.006** |
0.607 |
0.028* |
SUPRAMARGINAL |
0.895 |
0.305 |
0.575 |
0.021* |
RECTUS |
0.331 |
0.001** |
0.548 |
0.016* |
Cingulum (cingulate) |
0.548 |
0.016* |
0.331 |
0.001** |
Cingulum (hippocampus) |
0.978 |
0.653 |
0.814 |
0.162 |
Despite some limitation, the results and the discussion are interesting and well written.
Answer: We thank the Reviewer for appreciation of our study.

Reviewer 2 Report
Comments and Suggestions for Authors
Title: ‘Age-related decline in brain myelination: quantitative macro-molecular proton fraction mapping, T2-FLAIR hyperintensity volume, and anti-myelin antibodies 7 years apart’
Summary:
Authors have investigated the longitudinal assessment of global and regional myelination of the normal adult human brain using the fast macromolecular fraction (MPF) mapping, along with age-related changes in white matter (WM) hyperintensities on FLAIR-MRI and the levels of anti-myelin autoantibodies in serum.
Comments:
The authors' findings, revealing a global decline in myelination with aging, notably in the frontal white matter, corpus callosum, and caudate nucleus, offer valuable insights. However, the manuscript requires meticulous attention to enhance clarity. The methodology lacks sufficient detail, employing multiple software tools without adequate explanation, and involving repetitive segmentations. The author explicitly mentions acquiring MT, T1, PD, FLAIR, and T2 data, which can be highly useful for comparisons and segmentation.
Examples: In lines 166-167: What was the reason for using ITKsnap to calculate volumes? FSL is a good tool for the same.
In lines 169-173: Why is the registration being repeated with ANTS? i.e local changes could be calculated with ROI analysis.
Please use skull striped images , in the Figure 1, as BET was used to do skull striping. There is no mention of Bias field corrections, also there is no mention of coil setup in the manuscript.
The data representation needs improvement. Given the calculation of deformation fields, a more intuitive understanding could be achieved through voxel-wise analysis alongside bar graphs.
Author Response
We thank the Reviewer for valuable comments and critical evaluation.
Reviewer 2
Open Review
Summary:
Authors have investigated the longitudinal assessment of global and regional myelination of the normal adult human brain using the fast macromolecular fraction (MPF) mapping, along with age-related changes in white matter (WM) hyperintensities on FLAIR-MRI and the levels of anti-myelin autoantibodies in serum.
Comments:
- The authors' findings, revealing a global decline in myelination with aging, notably in the frontal white matter, corpus callosum, and caudate nucleus, offer valuable insights. However, the manuscript requires meticulous attention to enhance clarity. The methodology lacks sufficient detail, employing multiple software tools without adequate explanation, and involving repetitive segmentations. The author explicitly mentions acquiring MT, T1, PD, FLAIR, and T2 data, which can be highly useful for comparisons and segmentation.
Answer: We clarified description of the usage of specific software tools for specific processing tasks in the Methods section.
- Examples: In lines 166-167: What was the reason for using ITKsnap to calculate volumes? FSL is a good tool for the same.
Answer: We agree with the reviewer that volumes can be calculated in FSL. The use of ITKsnap is just a matter of personal convenience for our operators. They used ITKsnap primarily for lesion segmentation and are more accustomed to ROI operations in this package. We tested the results of ROI statistics using both packages and found that the voxel volumes calculated using FSL are exactly the same as the volumes calculated in ITK-snap.
- In lines 169-173: Why is the registration being repeated with ANTS? i.e local changes could be calculated with ROI analysis.
Answer: We have used standard-sized ROI placement previously in our animal studies (Khodanovich et al. 2017) and manual delineation of the basal ganglia by two operators in our human studies (Yarnykh et al. 2018). This experience showed that manual delineation can only be correctly applied to structures with clearly distinguishable boundaries, such as the basal ganglia. Manual delineation of structures with less contrast boundaries, such as white matter pathways, is highly operator-dependent and may lead to errors and inaccuracies. To avoid this issue, we have chosen an operator-independent image processing approach based on registration of the atlas template to individual MPF maps. Additionally, the accuracy of registration of the atlas segmentation to individual MPF maps was visually inspected in each case, and no subjects that would require manual correction were identified.
- Please use skull striped images, in the Figure 1, as BET was used to do skull striping. There is no mention of Bias field corrections, also there is no mention of coil setup in the manuscript.
Answer: We replaced all images with their skull-stripped versions. MPF maps are inherently insensitive to the coil reception profile. Their sensitivity to B1 inhomogeneity is very minor and practically negligible at 1.5T. For this reason, bias field correction was not used. We added this clarification in the Methods. Information about the coil also was added (please see answer to Reviewer 1).
- The data representation needs improvement. Given the calculation of deformation fields, a more intuitive understanding could be achieved through voxel-wise analysis alongside bar graphs.
Answer: We added voxel-wise maps of the structures with significant changes superimposed into MPF maps. Please see Fig. 5 for detail.

Reviewer 3 Report
Comments and Suggestions for Authors
Ref.: biomedicines-2708243
The authors studied age-related changes in brain myelination using fast macromolecular fraction (MPF) mapping in 11 healthy subjects. They observed a statistically significant decrease of MPF globally, as well as in many (48/82) studied regions, especially in frontal white matter, genu of corpus callosum and caudate nucleus. This age-related decrease was associated with an increase in serum antibodies against myelin basic protein.
Given that a gradual decrease of myelination occurs with age and is probably related with cognitive decline, such studies are important especially for vascular cognitive decline. In fact the present study tested MPF as a biomarker of myelin and showed that it could perform better than white matter hyperintensities in conventional FLAIR.
The study is well conducted and data well presented. Two minor points:
(a) What is GM? Please explain the 1st time it appears in the text.
(b) It would be interesting to study the cognitive performance of the participants. They are healthy, but could very sensitive tests especially of frontal function reveal subtle changes 7 years later? Of course such data are not available, but please add a short comment in the discussion section.
Author Response
We thank the Reviewer for valuable comments and critical evaluation
Reviewer 3
The authors studied age-related changes in brain myelination using fast macromolecular fraction (MPF) mapping in 11 healthy subjects. They observed a statistically significant decrease of MPF globally, as well as in many (48/82) studied regions, especially in frontal white matter, genu of corpus callosum and caudate nucleus. This age-related decrease was associated with an increase in serum antibodies against myelin basic protein.
Given that a gradual decrease of myelination occurs with age and is probably related with cognitive decline, such studies are important especially for vascular cognitive decline. In fact the present study tested MPF as a biomarker of myelin and showed that it could perform better than white matter hyperintensities in conventional FLAIR.
The study is well conducted and data well presented. Two minor points:
(a) What is GM? Please explain the 1st time it appears in the text.
Answer: We thank the reviewer for the attention to the text of the manuscript and appreciation of our study. Corrected: grey matter (GM).
(b) It would be interesting to study the cognitive performance of the participants. They are healthy, but could very sensitive tests especially of frontal function reveal subtle changes 7 years later? Of course such data are not available, but please add a short comment in the discussion section.
Answer: We thank the Reviewer for this important suggestion. The following paragraph have added to the discussion section:
Since many studies consider WM loss as the basis of cognitive decline in normal aging (Kohama et al. 2012; O’Sullivan et al. 2001), it would be of great interest to investigate the association of demyelination with performance on cognitive tests. Future studies will define the contribution of demyelination of specific brain structures with age-related decline of cognitive abilities.

Round 2
Reviewer 1 Report
Comments and Suggestions for Authors
The title should be modified, adding "preliminary findings"or "preliminary study" . Indeed, the sample size and the cost of the study that did not allow the use of an additional sample, for me are important issues.
Author Response
The title should be modified, adding "preliminary findings"or "preliminary study" . Indeed, the sample size and the cost of the study that did not allow the use of an additional sample, for me are important issues.
Response: We appreciate the Reviewer's suggestion and agree that it would be reasonable to highlight the preliminary nature of our findings. However, the title is already quite long, and any additional words would make it unnecessarily awkward and difficult to read. For this reason, we have chosen to modify the abstract and characterized this study as a preliminary one in the second sentence.

Reviewer 2 Report
Comments and Suggestions for Authors
Feedback: The document has undergone substantial enhancements and appears to be prepared for publication.
Comments: To be consistent, please add error bars in fig 2 and 3 in both directions ie +/-SD.
In Fig 4, please add a scale to the level of significance map, i.e., from 0.05 to 0.0001
Author Response
Feedback: The document has undergone substantial enhancements and appears to be prepared for publication.
Comments:
- To be consistent, please add error bars in fig 2 and 3 in both directions ie +/-SD.
Response: We have modified Figures 2 and 3 according to the Reviewer's suggestions..
- In Fig 4, please add a scale to the level of significance map, i.e., from 0.05 to 0.0001
Response: We have added significance level to Figures 4 according to the Reviewer's suggestions.

Round 3
Reviewer 1 Report
Comments and Suggestions for Authors
The authors addressed my last concern.